# Dynamic Evapotranspiration Alters Hyporheic Flow and Residence Times in the Intrameander Zone

**James Kruegler [1],\*** , **Jesus Gomez-Velez [2]**, **Laura K. Lautz [3]** and **Theodore A. Endreny [4]**

1   The Davey Institute, 1500 North Mantua Street, Kent, OH 4424, USA
2   Department of Civil and Environmental Engineering, Vanderbilt University, 400 24th Avenue South, 267 Jacobs Hall, Nashville, TN 37212, USA; jesus.gomezvelez@vanderbilt.edu
3   Department of Earth Sciences, Syracuse University, 204 Heroy Geology Laboratory, Syracuse, NY 13244, USA; lklautz@syr.edu
4   Department of Environmental Resources Engineering, State University of New York, College of Environmental Science and Forestry, 1 Forestry Drive, 402 Baker Labs, Syracuse, NY 13210, USA; te@esf.edu
\*   Correspondence: JKruegler@gmail.com

**Abstract:** Hyporheic zones (HZs) influence biogeochemistry at the local reach scale with potential implication for water quality at the large catchment scale. The characteristics of the HZs (e.g., area, flux rates, and residence times) change in response to channel and aquifer physical properties, as well as to transient perturbations in the stream–aquifer system such as floods and groundwater withdraws due to evapotranspiration (ET) and pumping. In this study, we use a numerical model to evaluate the effects of transient near-stream evapotranspiration (ET) on the area, exchange flux, and residence time (RT) of sinuosity-induced HZs modulated by regional groundwater flow (RGF). We found that the ET fluxes (up to 80 mm/day) consistently increased HZ area and exchange flux, and only increased RTs when the intensity of regional groundwater flow was low. Relative to simulations without ET, scenarios with active ET had more than double HZ area and exchange flux and about 20% longer residence times (as measured by the median of the residence time distribution). Our model simulations show that the drawdown induced by riparian ET increases the net flux of water from the stream to the nearby aquifer, consistent with field observations. The results also suggest that, along with ET intensity, the magnitude of the HZ response is influenced by the modulating effect of both gaining and losing RGF and the sensitivity of the aquifer to daily cycles of ET withdrawal. This work highlights the importance of representing near-stream ET when modeling sinuosity-induced hyporheic zones, as well as the importance of including riparian vegetation in efforts to restore the ecosystem functions of streams.

**Keywords:** hyporheic zone; hyporheic exchange; evapotranspiration; groundwater modeling; riparian vegetation

## 1. Introduction

The hyporheic zone (HZ) plays a crucial role in basic ecosystem functions in riparian corridors, being the region of an aquifer where there is some degree of mixing between stream water and groundwater. Focusing at the local reach scale, the exchange of water, solutes, and biota to and from the stream gives the HZ unique hydrodynamic and chemical properties [1], as well as a rich diversity of microbial communities [2]. Hyporheic exchange allows solutes carried by a stream to temporarily reside in the geochemically and microbially active streambed and banks [3], which can create the microfauna and solute residence times (RTs) necessary for critical biogeochemical transformation [4]—for example,

retention of nutrients [5] and metals [6]. Thus, hyporheic exchange influences major environmental engineering problems such as degradation of water quality [7], stream restoration [8], and the integrity of riparian ecosystems [9].

Spatiotemporal changes of the HZ's characteristics such as area, fluxes, and residence times can have a significant impact on its potential for biogeochemical transformation. For example, changes in the areal extent of the hyporheic zone can dictate the location of hotspots for biogeochemical transformations [10,11]. Similarly, changes in hyporheic flux strongly constrain the mass and spatial distribution of reactants available for transformations within the hyporheic zone. Lastly, hyporheic residence times can serve as a proxy for the likelihood that a solute will be consumed during a biogeochemical reaction (e.g., denitrification), and therefore to quantify the HZ's biogeochemical potential [6,12]. This can be done by comparing the HZ's RTs with a characteristic timescale for the reaction of interest, typically defined as the reciprocal of a reaction rate constant [13,14].

In general, the hyporheic zone hydrodynamics and its associated characteristics are defined by the porous media properties (e.g., permeability [15], porosity, specific yield, and dispersivity [14]); modulators such as regional groundwater flow [16], pumping [17], and evapotranspiration fluxes [18]; and drivers such as pressure gradients induced by interactions with geomorphic features (e.g., bedforms and meanders) [11,19–22]. In this work, we focus on the case of lateral hyporheic exchange driven by channel sinuosity (exchange between the channel and its banks) [11,19,22,23] as it is modulated by regional groundwater flow and riparian evapotranspiration.

Near-stream vegetation (e.g., phreatophytes) and diel cycles of radiation influence the hydrodynamics of the HZ though the combined effect of transpiration (root uptake) and evaporation of surface water, soil water, and groundwater within the alluvial valley. The evapotranspiration process results in a diurnal cycle of alluvial valley water withdraw, where the local water table declines throughout the day and recovers at night [24]. Lowering of the water table induces additional hydraulic gradients from the channel to the alluvial valley, potentially enhancing the hyporheic exchange process [23,25]. Patterns of fluctuations similar to the riparian ET diel cycles are commonly observed in near-stream well hydrographs [26] and stream discharge measurements [27]. These fluctuations have been attributed directly to evapotranspiration (ET) in areas where other potential perturbations on water table fluctuations (e.g., well pumping and barometric pumping) were considered negligible [24,28].

Despite the effects riparian ET has on fundamental aspects of the HZ's hydrodynamics, relatively few numerical models of the HZ account for any amount of ET, and fewer have attempted to quantify the relationships between ET and metrics such as HZ area, exchange flux, and RTs. With this in mind, the goal of this work is to identify the magnitudes of ET withdrawal necessary to alter major characteristics of lateral reach scale hyporheic exchange, across a range of regional groundwater conditions. This goal is central to our ongoing effort to identify and maximize the potential for environmentally beneficial hyporheic activity along river corridors.

To meet our goal, we modeled and evaluated the effects of transient near-stream ET in a numerical model for lateral reach scale hyporheic exchange. To put the results in proper context, the effects of ET were compared with results from previous modeling studies (e.g., [11,14,19]) that have simulated a range of stream morphologies and ambient groundwater conditions to identify maximum values for HZ area, hyporheic exchange flux, and RTs.

Using a finite-element model, Cardenas [11,19,20] explored the relationship between sinuosity-driven hyporheic exchange, channel sinuosity, and regional groundwater flow. Their simulations showed that channel morphology, as represented by sinuosity, exerts a dominant control in the shape and fluxes within the HZ. The exchange through homogeneous meanders is characterized by broad power law residence time distributions (RTDs) extending across several orders of magnitude, from minutes to decades. This is an important finding that suggests prolonged effects of hyporheic exchange on the biogeochemical transformation potential at scales that range from the reach to the watershed [20]. Specifically, gaining or losing regional groundwater flow (RGF) conditions compress the HZs toward

meander apexes and shorten residence times [11]. As sinuosity increases, the sensitivity to RGF decreases and steeper transverse regional water table gradients are needed to compress the HZ [11].

Boano et al. [13] used a similar modeling framework to make a direct connection between hyporheic flow, RTs, and biogeochemical transformation in the intra-meander zone. Hyporheic flow was simulated in meanders with a wide range of channel sinuosities, producing flow paths and residence time distributions that vary widely between scenarios. Then, a reactive solute transport model was used to simulate a sequence of redox reactions as a steady supply of dissolved organic carbon was supplied by the channel to the HZ. These biogeochemical simulations show the tight connection between residence time distributions and the potential for denitrification. In particular, Boano et al. [13] highlighted the usefulness of comparing threshold biogeochemical timescales (associated to the reaction of interest) with the HZ's residence times in order to quantify the effects of stream morphology on the biogeochemical potential of the HZ. Expanding upon this work, Gomez et al. [14] used the biogeochemical model from Boano et al. [13] with the numerical flow model from Cardenas [11,19] to establish numerical relationships between sinuosity, water table gradient, hydraulic conductivity, aquifer dispersivity, and RTDs. Keeping aquifer dispersivity constant, changes in the other aquifer and morphological parameters either stretched or compressed the RTD for each simulation. By comparing those RTDs with biogeochemical timescales [13], each set of control parameters could be used to classify a meander as a net sink or source of nitrate.

In an investigation of the relationship between riparian ET and streamflow diel cycles, Wondzell et al. [23] examined ET-induced fluctuations in streamflow at the mouth of a watershed. When compared to data from previous stream-tracer experiments performed throughout the watershed, they concluded that the combination of ET signals from local and watershed scales produced a nonlinear relationship between ET and streamflow. The study results also suggested that some observed water table fluctuations were in response to ET perturbation effects being transported along hyporheic flow paths. In other words, the effects of ET on hyporheic exchange can persist in the HZ beyond the time scale associated with the daily cycle. This is consistent with the memory effects found by Gomez-Velez et al. [29] in the context of flood event perturbations.

Efforts to characterize the effects of transient model boundaries on HZ exchanges and RTs, and in turn on the biogeochemical potential of HZs, have been limited and remain inconclusive. Gomez-Velez et al. [29] explored the effect of dynamic (or transient) perturbations due to flood events on lateral hyporheic exchange. These perturbations result in a wide variation in exchange fluxes and residence times. The variable perturbation in the model took the form of transient stream discharge, representing flood pulses of varying duration and intensity. Gomez and Wilson [30] drew similar conclusions after modeling transient flow in a generic flow system, where large enough fluctuations in the model perturbations create entirely new modes in the resulting distributions of residence time distributions. The results of Larsen et al. [31], however, suggest that dynamic perturbations from flood pulses and ET affect HZs on different spatial and temporal scales. The study ran multi-year simulations of hyporheic exchange, all with seasonal-timescale flood pulses and some with ET changing on a daily cycle. The ET boundary had different effects on HZ metrics at different temporal scales: on a monthly timescale, hyporheic exchange fluxes increased by several orders of magnitude; six-year mean residence times decreased by over half compared to no ET. The authors concluded that seasonal flood pulses produced separate characteristic RTDs from the ET effects taking place at shorter timescales.

The limited studies of transient conditions in hyporheic models leave some aspects of the research in a state of debate. Characteristic hyporheic RTs have been used to estimate the potential for biogeochemical transformation in HZs [13,32], but many studies have reported multimodal RTDs are inadequately described by a single characteristic RT [20,21]. Wondzell et al. [23] and Larsen et al. [31] suggest that transient model conditions produce complex multimodal distributions because they affect HZ transport and storage dynamics on multiple timescales. This is particularly important given than biogeochemical timescales for oxygen and nitrate consumption in meanders can vary over nine orders of magnitude from $10^{-2}$ to $10^6$ days [14].

## 2. Materials and Methods

### 2.1. Conceptual Model

We used MODFLOW-2005 to build a finite-difference model of the hyporheic exchange process modulated by ET and regional groundwater flow (RGF) (Figure 1). Our conceptual model is based on the model previously proposed by Cardenas [11,19] and revisited by Gomez et al. [14] and Gomez-Velez et al. [29]. The generic domain represents one side of an alluvial valley overlying an impermeable surface. The valley is constrained on the *x-y* plane by the outer edge of the valley on one side and the valley's stream on the opposite side. The left and right (upstream and downstream, respectively) domain edges are periodic boundaries, i.e., they are boundaries beyond which the stream meanders repeat periodically and indefinitely. The model domain is vertically-integrated in a 2 m-thick layer, the stream fully penetrates that layer, and the alluvial aquifer is homogenous and isotropic. The stream channel is sinusoidal, with fixed head values varying linearly along the arc-length of the channel.

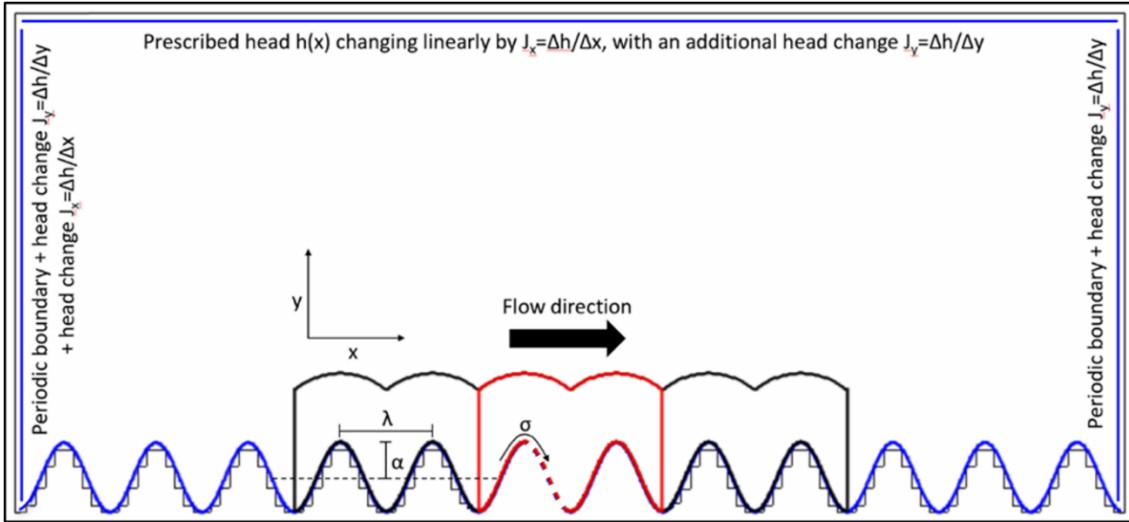

**Figure 1.** Overview of the finite-element model domain, with all boundary conditions labeled. The near-stream area outlined in black, encompassing the smaller area outlined in red, is the full extent of the vegetated riparian area, simulated with MODFLOW's EVT boundary condition. The area outlined in red is the area of interest pictured in proceeding figures. The dashed red line corresponds to the segment of the stream–aquifer interface along which MODPATH particles were released. Blue lines correspond to constant head (CHD) boundaries.

All four edges of the model domain were represented by the MODFLOW Time-Variant Specified Head package (constant head, or CHD). These constant heads varied linearly along a single border to produce the desired mean head gradients in the *x*- and *y*-directions and thereby control mean water table configurations before the addition of ET [11]. The near-stream ET was generated by MODFLOW's Evapotranspiration package (EVT) and the riparian vegetation was represented as a band of EVT cells running adjacent and parallel to the stream. The EVT cells ran on a daily cycle of hourly pumping rates (Figure 2), to reflect the observed sub-daily effects of solar radiation on groundwater ET [27]. Transport of stream water through the aquifer was modeled both as a conservative solute using the Modular Three-Dimensional Multispecies Transport Model (MT3DMS) [33], and as a series of discrete particles using the particle-tracking program MODPATH [34]. Observations from these transport software packages were used to calculate different metrics (described in Section 2.3) and did not interact or exchange data.

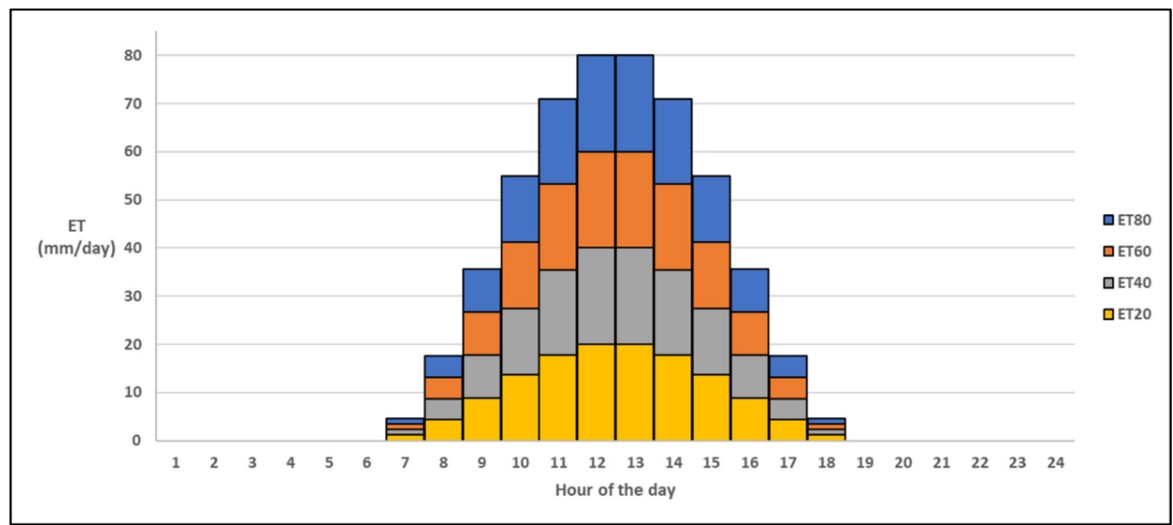

**Figure 2.** Daily schedules of evapotranspiration pumping rates in the vegetated zones of the model. Schedules are color-coded according to the maximum pumping rate (mm/day) achieved in each schedule, and labeled in shorthand (e.g., ET80 = maximum rate of 80 mm/day).

To represent an infinitely long stream and vegetated zone, the domain was expanded beyond the central 2 wavelengths of the stream that make up the representative area of interest (Figure 1, in red). Measurements of all metrics aside from HZ area come exclusively from this area. The top, right, and left edges of the domain (in plan view, Figure 1) were expanded until the total steady-state fluxes entering or leaving those edges during active ET (80 mm/day, the maximum withdrawal rate simulated) were within 5 percent of the same fluxes from simulations without active ET (see the description of scenarios below). Then, wavelengths with active ET zones were added upstream and downstream of the area of interest until, under transient conditions with daily ET cycles, drawdown and head within the area of interest experienced a maximum change of <5% due to the additional ET wavelengths.

The top boundary of the vegetated zone ET was established in a way that places it at least 30 m away from the stream at all points along the stream. This was done to maintain a more consistent thickness to the vegetated zone ET, in sync with the sine wave of the stream, than would have been produced with a straight line running along the *x*-axis of the domain. The 30 m vegetation width is consistent with recommendations from the USDA [35] for riparian buffers. A more detailed description of how this boundary was created in ModelMuse [36] is provided in the Supplementary Materials.

Some aspects of the model were chosen with the goal of avoiding numerical instability and water budget imbalances in the finite-difference solution. The model's *x*-*y* grid was initially given a regular cell size of 4 × 4 m, but the cells in the area of interest were refined to 0.4 × 0.4 m, with several rows and columns of transitional cells in-between. This gives the model over 47,000 active cells total and over 22,000 active cells in the area of interest alone. The MODFLOW Geometric Multigrid (GMG) solver package was chosen because it kept water budget discrepancies below 0.1% for all steps in all simulations. All solver packages solve the same equations for groundwater flow, but with different algorithms that allow some to arrive at satisfactory solutions when others will not [37].

*2.2. Modeling Scenarios*

Five different scenarios for ET withdrawal rates were applied, based on the daily maximum ET rate ($ET_{max}$): 0, 20, 40, 60, and 80 mm/day. These scenarios will be abbreviated from here onward as ET0, ET20, ET40, ET60, and ET80, respectively. These values cover the full range of ET activity found in the literature [26,27,38,39]; but it should be noted the ET withdrawal rates produced in the ET80 scenarios will be impossible for phreatophytes to achieve in most climate zones. The mean head gradient of the aquifer in the *y*-direction ($J_y$) was also altered between simulations to produce

different scenarios of ambient groundwater flow. The ratio of the regional groundwater flow gradient in the *y*-direction ($J_y$) to the one in the *x*-direction ($J_x$) is given by $J_{y/x} = J_y/J_x$. This ratio represents the magnitude of the regional groundwater flux constraining the hyporheic exchange, and it is positive for gaining conditions, negative for losing conditions, and zero for neutral conditions. We explored five different scenarios for $J_{y/x}$, with values of 2, 1, 0, −1, and −2. These scenarios will be abbreviated from here onward as J + 2, J + 1, J0, J − 1, and J − 2, respectively. Most model parameters were kept constant across all simulations, to reduce the total number of simulations and simplify the analysis of the parameters that were changed. The main model constants can be found in Table 1.

**Table 1.** Constant values for major geomorphic and aquifer parameters.

| Parameter | Symbol and Units | Constant Value |
|---|---|---|
| Sinuosity | S, (-) | 1.87 |
| Wavelength | λ, m | 40 |
| Down-valley water table gradient | $J_x$, (-) | 0.00125 |
| Hydraulic conductivity | K, m/h | 3.5 |
| Porosity | φ, (-) | 0.25 |
| Specific yield | $S_y$, (-) | 0.20 |
| Longitudinal dispersivity | $\alpha_L$, m | 10 |

The EVT cells were set with extinction depths at the bottom of the model layer (2 m) and withdrawal coming from the top of the model layer. As a result, the real withdrawal rate of an active EVT cell was a fraction of the maximum possible withdrawal rate of a given time step (the hly values of pumping rates in Figure 2), with the fraction being proportional to the head in the cell:

$$ET_R = ET_P \times (h/2) \tag{1}$$

where $ET_R$ is the real withdrawal rate (mm/day) of a cell, $ET_P$ is the maximum potential withdrawal rate (mm/day) for that time step, and *h* is the head (m) in the cell. An extinction depth of 2 m is a plausible value for riparian phreatophytes, based on ranges of values produced in previous modeling studies [40–42].

Each simulation was run with one steady-state flow step, one steady-state solute transport step, and then 744 steps (31 days) of hly ET and transient solute transport (Table 2). Representative measurements of stream water concentrations and flow terms needed for area and flux metrics were taken from the 31st day of the transient simulation. The particle tracking used for residence time calculations ran beyond the first 31 days for reasons described below. Post-processing of model data was completed with a combination of tools from ModelMuse and scripts written in R [43]. For more detailed explanations of the model setup and execution, see the Supplementary Materials.

**Table 2.** The changing status and activity of the MODFLOW, MT3DMS, and MODPATH software through a simulation's time steps.

| Start Time Step | End Time Step | MODFLOW | MT3DMS | MODPATH |
|---|---|---|---|---|
| −1 | 0 | Steady-state | Inactive | Inactive |
| 0 | 1 | Transient | Steady-state | Inactive |
| 1 | 720 | Transient | Transient | Inactive |
| 720 | 721 | Transient | Transient | Particles released |
| 721 | 744 | Transient | Transient | Particles travel |
| 744 | variable | Transient | Transient | Particles travel |

*2.3. Characterization of the Hyporheic Exchange*

The areal extent of the HZ was evaluated using a geochemical definition proposed by Triska et al. [44] and used by Gomez-Velez et al. [29], where the HZ is the area within the aquifer

composed of more than 50% stream water. The geochemical definition of the hyporheic zone has been identified in previous research [13,14,32] as an effective method of predicting the potential for biogeochemical reactions. At the beginning of each simulation the stream water, with an initial concentration of 1 g/m$^3$, was released from the upstream half of the central wavelength of the stream into the aquifer with an initial concentration of 0. At the end of each simulation, each cell with a final stream water concentration greater than 0.5 g/m$^3$ was added to the HZ area ($A_{HZ}$). The percent increases in $A_{HZ}$ relative to the corresponding ET0 scenarios (i.e., with the same $J_{y/x}$) were also calculated.

The active ET in this model acted as a sink distributed across a wide band of cells, some of them up to 30 m away from the stream. Two flux metrics were developed to provide a look at fluxes throughout that distributed sink as well as fluxes exchanged along the stream–aquifer interface. The first metric was a net flux term in the $y$-direction ($Q_y$), evaluated cell by cell with positive terms flowing toward the stream. Cross-sectional profiles of $Q_y$ values in the central wavelength were plotted for direct comparison between simulations.

The second flux metric was a normalized dimensionless flux ($F$) describing exchange along the stream–aquifer interface. On the 31st day of each simulation, a daily average flux was calculated for the stream–aquifer interface of the central two stream wavelengths using ZoneBudget. This daily average flux was made dimensionless ($Q^*$):

$$Q^* = Q_{ave}/(Kh_c{}^2) \tag{2}$$

where $Q_{ave}$ is the calculated daily average flux (m$^3$/h), $K$ is hydraulic conductivity (m/h), and $h_c$ is a characteristic head value (1 m). Then the $Q^*$ terms were normalized to the ET0 simulation with the same $J_{y/x}$:

$$F = Q^*/Q_0 \tag{3}$$

where $F$ is the normalized dimensionless flux and $Q_0$ is the dimensionless daily average flux from the respective ET0 simulation.

Because the model was transient and there were changes in model storage, aquifer residence times (RTs) were estimated with a Lagrangian approach, which involved tracking individual particles through time as they traveled through the area of interest. Along the upstream half of the central wavelength, a particle was placed on every cell face along the stream–aquifer interface at a depth of 1m, for a total of 125 particles. Before release, MODFLOW was run with a wind-up time that replicated the conditions imposed on the model for the other metrics: one steady-state flow step with no ET and then 720 transient time steps (30 days) with daily cycles of ET withdrawal. After this wind-up time, the particles were released and MODFLOW was run with the same daily ET cycle until all particles had exited the area of interest. Sets of particles were released during the 1st, 6th, and 12th hours of the 31st day, to explore the sensitivity of the metrics described below to particle release time.

The RT (h) of each particle was weighted according to the volumetric fluxes of stream water entering the HZ at the time the particles were released, along the section of the stream–aquifer interface where particles were released. Flux-weighted RT values ($RT_{FW}$, h) are defined as:

$$RT_{FW} = RT^*(Q_P/Q_T) \tag{4}$$

where $Q_P$ is the flux represented by a particle (m$^3$/h) entering the aquifer at the cell face corresponding to the starting location of that particle, and $Q_T$ is the total flux of stream water entering the aquifer (m$^3$/h) along all cell faces on the upstream half of the central meander bend. Flux-weighted RTs were divided by a characteristic timescale of 24 h to produce dimensionless RT ($RT^*$) values reflecting the 24 h-long cycle of ET withdrawal. For each simulation, these dimensionless values were arranged in histograms, cumulative distribution functions (CDFs), and plotted as a function of dimensionless arc-length distance ($\sigma^*$):

$$\sigma^* = \sigma/\sigma_T \tag{5}$$

where $\sigma$ is the distance (m) along the arc-length of the upstream half of the central meander bend, increasing when moving downstream, and $\sigma_T$ is the total arc-length of the upstream half of the bend. Median $RT^*$ values from all scenarios were plotted once as a function of $J_{y/x}$ and $ET_{max}$, and then median $RT^*$ values from active ET scenarios were normalized to their respective ET0 scenarios and plotted again.

To provide mechanistic explanations of how hydraulic head influences the above metrics, contour maps of drawdown and head were interpreted in terms of how they affected the velocity and direction of groundwater flow paths relative to ET0 scenarios. Snapshots of drawdown and head contours were taken on the 31st day of the five ET80 simulations. On the head contour maps, the general orientation of the contours was estimated by drawing a straight line between two points: the top apex of the left meander bend in the area of interest and the cell with the lowest head value in the area of interest (Figure 1). The orientation of this line was recorded as an azimuth, or degrees clockwise from north. The average head gradients of these lines were also calculated. For each of the five simulations, these orientations and gradients were compared between the 6th and 12th h snapshots. These metrics were used to compare conditions just before the daily cycle of active ET with conditions while $ET = ET_{max}$. The patterns in the drawdown maps were not quantified but used to support conclusions drawn from other metrics.

Time sensitivity was quantified using earlier methods. The ET in this study was expected to affect the stream–HZ–alluvial aquifer system similar to the use of flooding events in Gomez-Velez et al. [29], in that dynamic ET rates would produce dynamic responses in all of the above metrics. The intensity of change in these responses depend on the sensitivity of the aquifer to the daily cycles of ET withdrawal, and this sensitivity was quantified with an adapted version of a hydraulic time constant ($t_h$) used in Gomez-Velez et al. [29]:

$$t_h = (S_y \lambda^2)/(Kb) \tag{6}$$

where $S_y$ is specific yield, $\lambda$ is meander wavelength, and $b$ is aquifer layer thickness. The hydraulic time constant was compared to the time scale of one cycle of ET withdrawal (24 h) to establish the relative importance of ET perturbations on the near-stream aquifer:

$$\Gamma = t_h/24 \tag{7}$$

If $\Gamma < 1$, the aquifer was considered insensitive and the water table configuration would likely not change in response to the daily cycle of ET withdrawal; if $\Gamma > 1$, the opposite was true.

## 3. Results

### 3.1. Hyporheic Zone Area

Compared to their base ET0 simulations, all regional groundwater flow (RGF) scenarios (intensity of RGF is proportional to $J_{y/x}$) showed relative increases in the hyporheic zone area ($A_{HZ}$) during simulations with active ET (Figure 3). The greatest relative increases were found in gaining scenarios ($J_{y/x} > 0$), where $A_{HZ}$ increased over 100% from ET0 to ET80. The strongest losing scenario ($J_{y/x} < 0$) had markedly smaller growth, with just over a 6% increase from ET0 to ET80.

Active ET simulations also showed altered HZ geometry. For gaining and neutral scenarios, the HZ expanded primarily along the $y$-axis of the domain. In losing scenarios, any growth or movement of the HZ was along the $x$-axis (see the fourth and fifth rows in Figure 3). After the steady-state transport step in each simulation, the shapes and sizes of the HZ did not change on daily timescales, and so only a single final snapshot was taken from each simulation.

### 3.2. Net Groundwater Flux

For the net flux in the $y$-direction ($Q_y$), gaining scenarios were the only scenarios to show any movement in the divide between positive terms representing net flux away from the stream and negative terms representing net flux toward the stream (Figure 4a,b top). Increasing ET rates, whether through the hly progression of one simulation or between simulations with different $ET_{max}$, corresponded to

the divide moving away from the stream and a greater fraction of the active ET zone showing a net flux away from the stream. Although this divide did not move in neutral and losing scenarios, these scenarios' profiles of discrete $Q_y$ values (Figure 4c–e, bottom) still showed greater flux away from the stream in scenarios and time steps with higher ET rates. This increased flux away from the stream could be seen in lower average values, lower minimum values, and lower values at the stream–aquifer interface (A' location on each profile). Of those three indicators, the values at the stream–aquifer interface may be the most crucial because they were direct measurements of exchange between stream water and hyporheic water. On average, when comparing ET0 scenarios to corresponding 12th h ET80 scenarios, $Q_y$ values at the stream–aquifer interface decreased by 117%, with a maximum difference of 11% between $J_{y/x}$ scenarios. All active ET scenarios displayed increased flux away from the stream regardless of $ET_{max}$, with the only difference being that smaller $ET_{max}$ corresponded to smaller $Q_y$ values. To simplify the snapshots the ET0 scenarios were only compared to ET80 scenarios. $Q_y$ values from the last 12 h of the daily ET curve were identical and symmetrical to those from the first 12 h, so only snapshots from the first 12 h are shown.

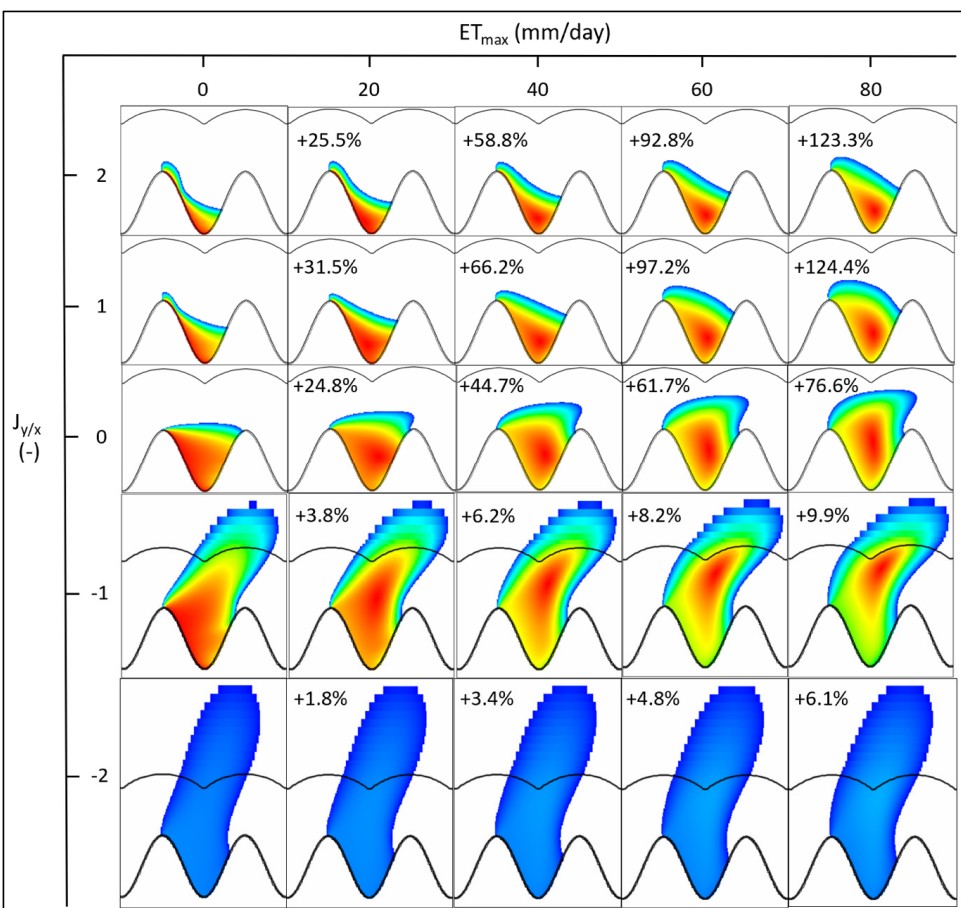

**Figure 3.** Maps of hyporheic zone area ($A_{HZ}$) as a function of maximum daily ET ($ET_{max}$) (columns) and regional groundwater flux ($J_{y/x}$; positive for gaining, negative for losing) (rows). Each map is a snapshot taken at the 12th h of the last day of the simulation. The percentages listed are the percent increases in $A_{HZ}$ relative to the corresponding ET0 scenarios (left column). The colors on each map correspond to concentrations of stream water that have entered the aquifer; the range of concentrations are different for each simulation and so the colors cannot be compared between maps. The colors follow a rainbow gradient where blue indicates lower concentrations and red indicates higher concentrations.

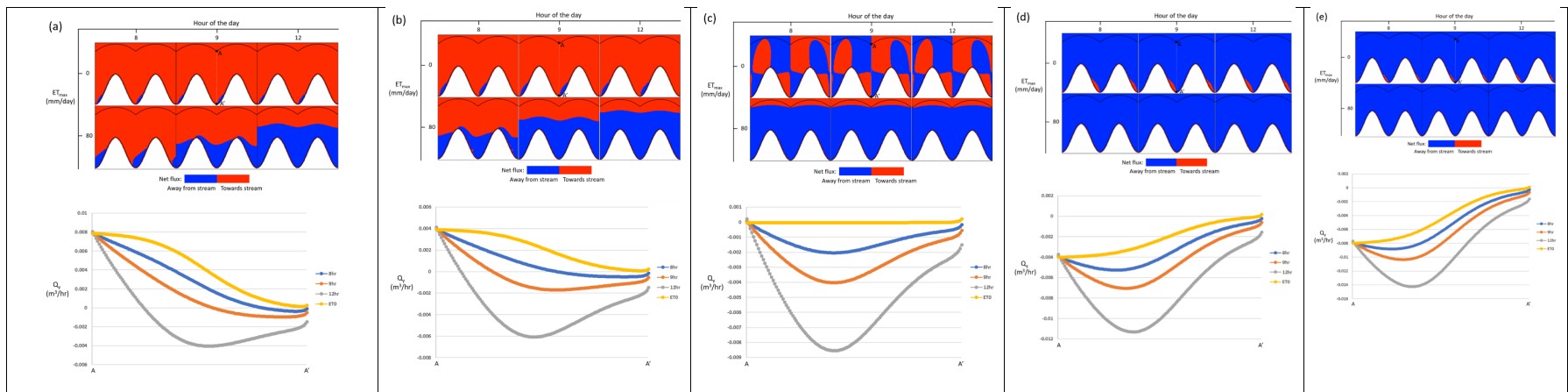

**Figure 4.** Snapshots of net $Q_y$ values at different h of the day (**top row**), as a function of maximum daily ET and regional groundwater flux ($J_{y/x}$; positive for gaining, negative for losing). Profiles of the above $Q_y$ (positive towards the stream) values along the middle of the vegetated area of interest (**bottom row**). Note that the no-ET scenario is unchanging with time and plotted only once on each profile. Columns correspond to simulations where the magnitude of $J_{y/x}$ was, respectively, (**a**) 2, (**b**) 1, (**c**) 0, (**d**) −1, and (**e**) −2.

Normalized dimensionless flux ($F$) increased across all $J_{y/x}$ scenarios when directly comparing ET0 simulations with their respective ET80 simulations (Figure 5). In other words, all ET80 simulations produced $F > 1$. For neutral and losing scenarios, all increases in $ET_{max}$ corresponded with increases in $F$, and ET80 $F$ values fall between 1.5 and 1.7. Gaining scenarios produced $F < 1$ at lower $ET_{max}$ and eventually $F > 1$ starting at ET60 for J + 1 and only at ET80 for J + 2. The lowest $F$ values, both overall (0.87) and on average (0.94), were produced in J + 2 scenarios.

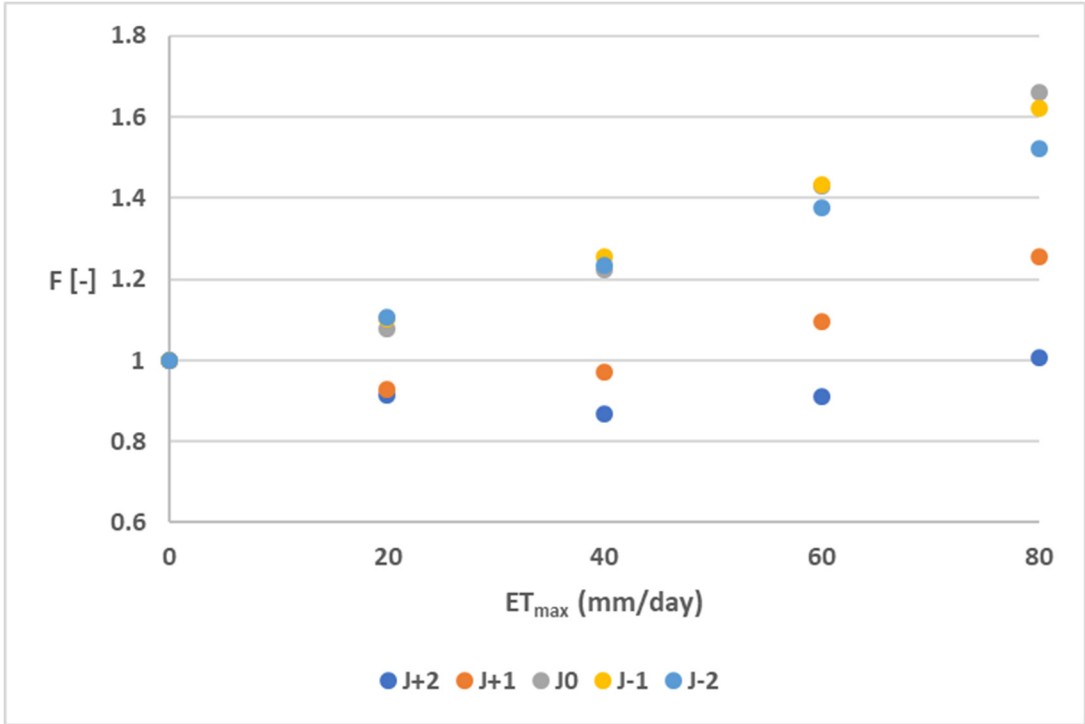

**Figure 5.** Normalized dimensionless flux ($F$) for each regional groundwater flux ($J_{y/x}$; positive for gaining, negative for losing) scenario, plotted as a function of $ET_{max}$.

### 3.3. Residence Time

For all simulations, the difference in median $RT^*$ between particle release times was less than 5%. The $RT^*$ values used for analysis came from the particles released during the 1st h of the 31st day of each simulation.

The residence time distributions (RTDs) displayed in the histograms and cumulative residence time distributions (CRTDs) in the CDFs developed unique shapes in response to both $J_{y/x}$ and $ET_{max}$. Gaining simulations displayed a strong early mode and long tail (Figure 6a,b), and this basic RTD shape did not change between ET0 and active ET scenarios. Neutral simulations produced a similar early mode distribution with no ET (Figure 6c, top), but with a less severe peak and a more gradual decline in the tail. With active ET, the early mode remained but a larger number of particles clustered around a range of high $RT^*$ values, eventually producing a smaller secondary mode in that range at the highest $ET_{max}$ (Figure 6c, bottom). Compared to ET0, median $RT^*$ and standard deviation more than doubled in ET80. Progressing from ET0 to ET80, losing scenarios (Figure 6d,e) began with large early modes that shifted to higher and higher $RT^*$ values, eventually settling close to the median $RT^*$. J − 2 scenarios (Figure 6e) began with several smaller secondary modes that shrank and disappeared as $ET_{max}$ increased. Active ET decreased standard deviations and compressed RTDs for all losing scenarios.

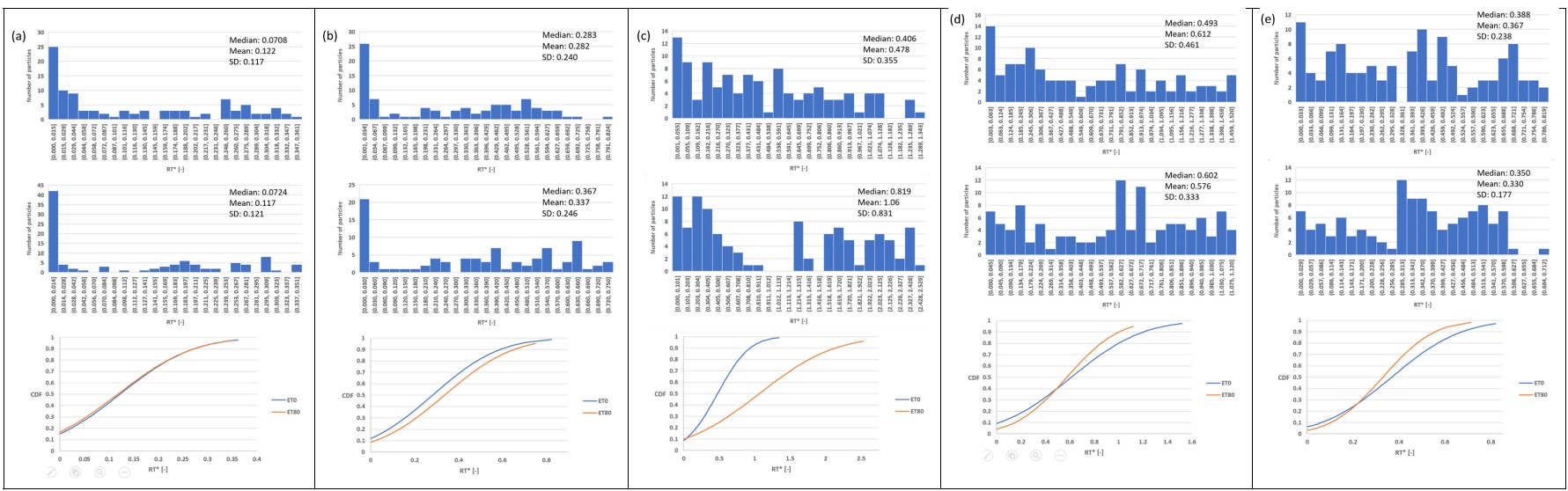

**Figure 6.** Residence time distributions (RTDs) from no-ET (ET0) scenarios (**top row**). RTDs from scenarios where maximum daily ET ($ET_{max}$) is 80 mm/day (**middle row**). Comparison of cumulative RTDs from ET0 and ET80 scenarios (**bottom row**). Columns correspond to simulations where the magnitude of regional groundwater flux ($J_{y/x}$; positive for gaining, negative for losing) was, respectively, (**a**) 2, (**b**) 1, (**c**) 0, (**d**) −1, and (**e**) −2.

Changes in median $RT^*$ as a response to $ET_{max}$ corresponded to the absolute magnitude of $J_{y/x}$. As $ET_{max}$ increased, intensely gaining and losing scenarios ($|J_{y/x}| = 2$) produced mostly decreases in median $RT^*$, and less intense and neutral scenarios ($|J_{y/x}| = \{0, 1\}$) produced mostly increases in median $RT^*$ (Figure 7). The largest changes in median $RT^*$ were over 20%: ET40 to ET60 and ET60 to ET80 in neutral scenarios produced >20% higher median $RT^*$, and ET20 to ET40 in strongly gaining (J + 2) scenarios resulted in >20% lower median $RT^*$.

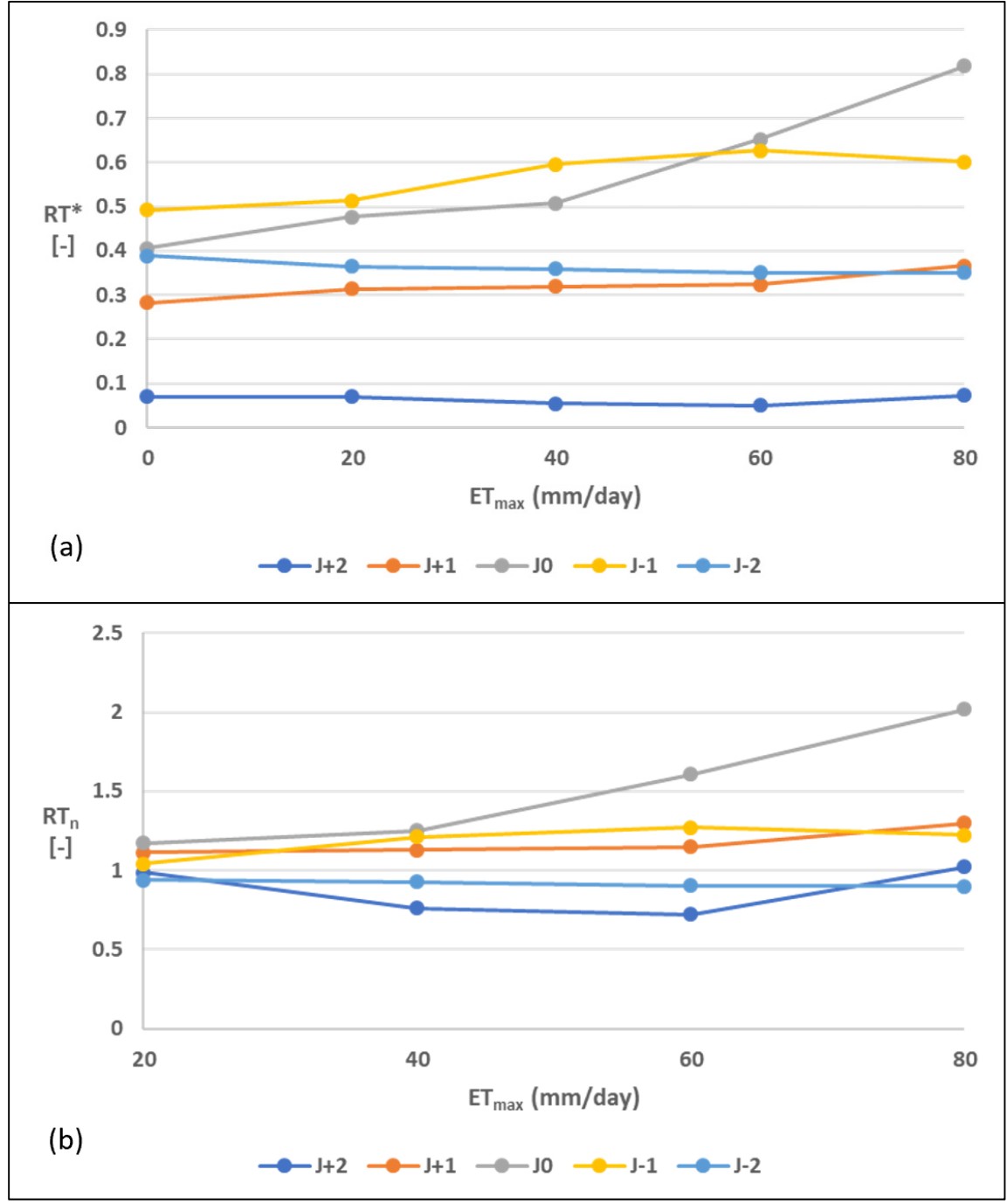

**Figure 7.** (**a**) Median dimensionless RT ($RT^*$) for all regional groundwater flux ($J_{y/x}$; positive for gaining, negative for losing) scenarios as a function of $ET_{max}$. (**b**) Median $RT^*$ values from active ET scenarios, normalized to their respective ET0 simulations.

Regional water table slope also affected the spatial distribution of RTs along the stream–aquifer interface. Gaining scenarios (Figure 8a,b) produced near-normal distributions of $RT^*$, with the highest values close to $\sigma^* = 0.5$ and the lowest values close to $\sigma^* = \{0, 1\}$. The greatest changes in $RT^*$ from ET0 to ET80 occurred near $\sigma^* = 0$, with decreases of over 90% in J + 2 and increases of over 2000% in J − 1. Neutral scenarios (Figure 8c) had comparatively flat spatial distributions, with slightly higher $RT^*$ closer to $\sigma^* = 0$. Nearly all $RT^*$ values increased from ET0 to ET80, with maximum increases of over 300% around $\sigma^* = 0.65$. With no ET, losing scenarios (Figure 8d,e) displayed two spatially distinct modes of $RT^*$, where average $RT^*$ was about four to five times higher where $\sigma^* < 0.5$ compared to $\sigma^* > 0.5$. At ET80, the high $RT^*$ modes dropped and the low $RT^*$ modes rose to produce a flat spatial distribution like the ones seen in neutral scenarios. The greatest changes in $RT^*$ were from increases in $RT^*$ where $\sigma^* < 0.5$; these increases were >300% in J − 1 and >200% in J − 2.

The point-to-point noise on the spatial distribution graphs is due to the fine-scale jagged edge of the stream–aquifer interface. This resulted in alternating groups of particles being initially placed on either side of the ideal, perfectly sinuous stream–aquifer interface being approximated by the model domain. This, in turn, produced RTs based on flow paths that are slightly too long or too short. To reduce some of this noise and make the spatial distributions more legible, the values plotted are moving five-point averages of the original $RT^*$ values.

### 3.4. Drawdown and Head

For all scenarios, the maximum observed drawdown was close to 0.1 m and centered on the top edge of the vegetated zone farthest from the stream (Figure 9), indicating the stream was the major source of recharge to the riparian aquifer. If the ET zone was drawing in a similar quantity of water from the alluvial valley, maximum drawdown values could be expected along the center of the ET zone, equidistant from the stream and alluvial valley.

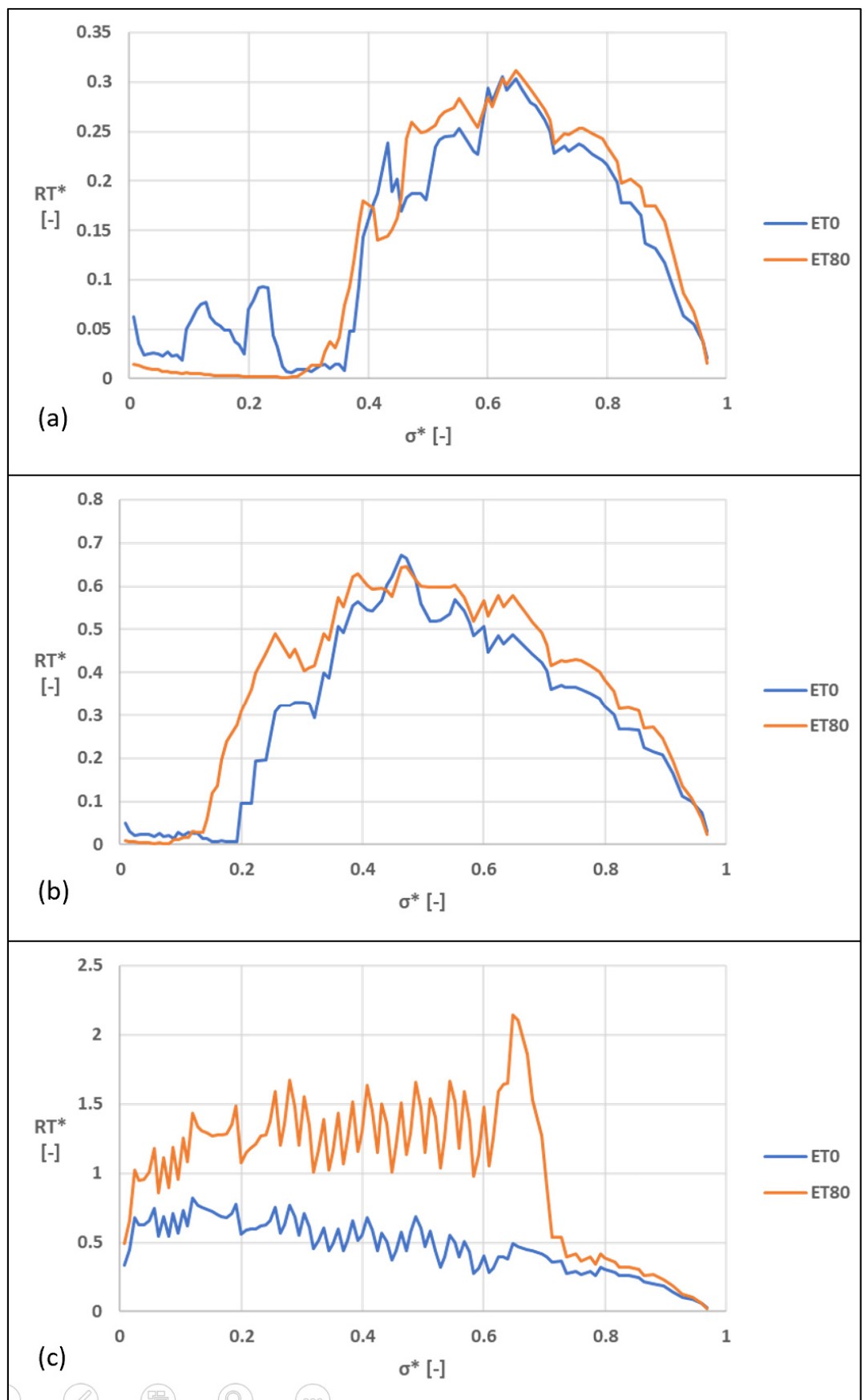

**Figure 8.** *Cont.*

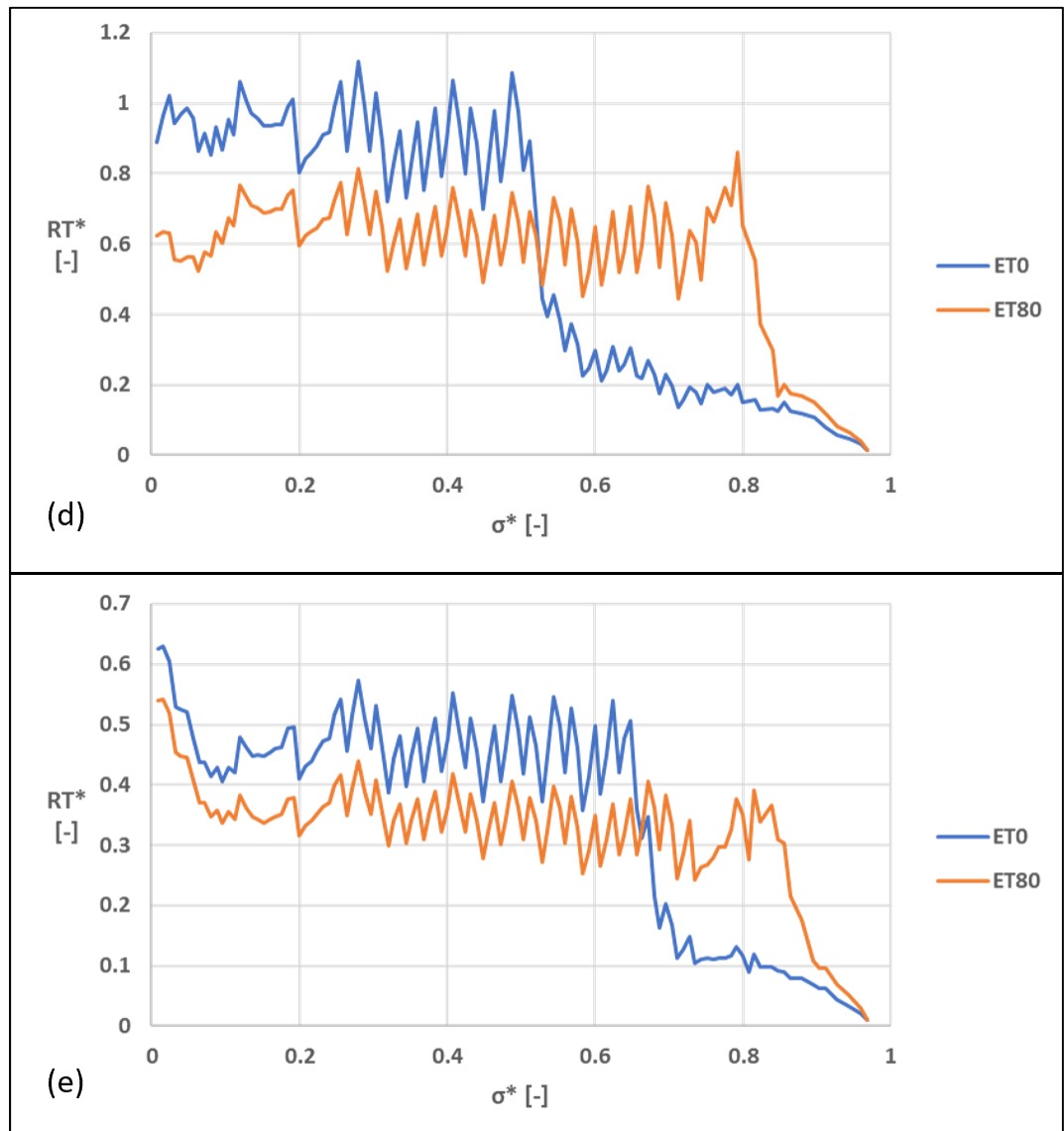

**Figure 8.** Moving 5-particle average *RT\** as a function of regional groundwater flux ($J_{y/x}$; positive for gaining, negative for losing) and initial particle placement along the upstream half of the central meander bend. Comparing ET0 and ET80 scenarios for (**a**) J + 2, (**b**) J + 1, (**c**) J0, (**d**) J − 1, and (**e**) J − 2.

For the head distributions (Figure 10), increased ET rates resulted in contour lines in the ET zones orienting themselves away from the stream and forming steeper gradients in some locations (see Table 3). Contour lines in gaining and neutral scenarios reoriented so the lowest head value in the area of interest moved away from the stream. In losing scenarios, the lowest head value in the area of interest did not reorient at all relative to the point placed at the meander apex. All scenarios produced a steeper head gradient when ET was active, with neutral and losing scenarios producing slightly higher changes in gradient.

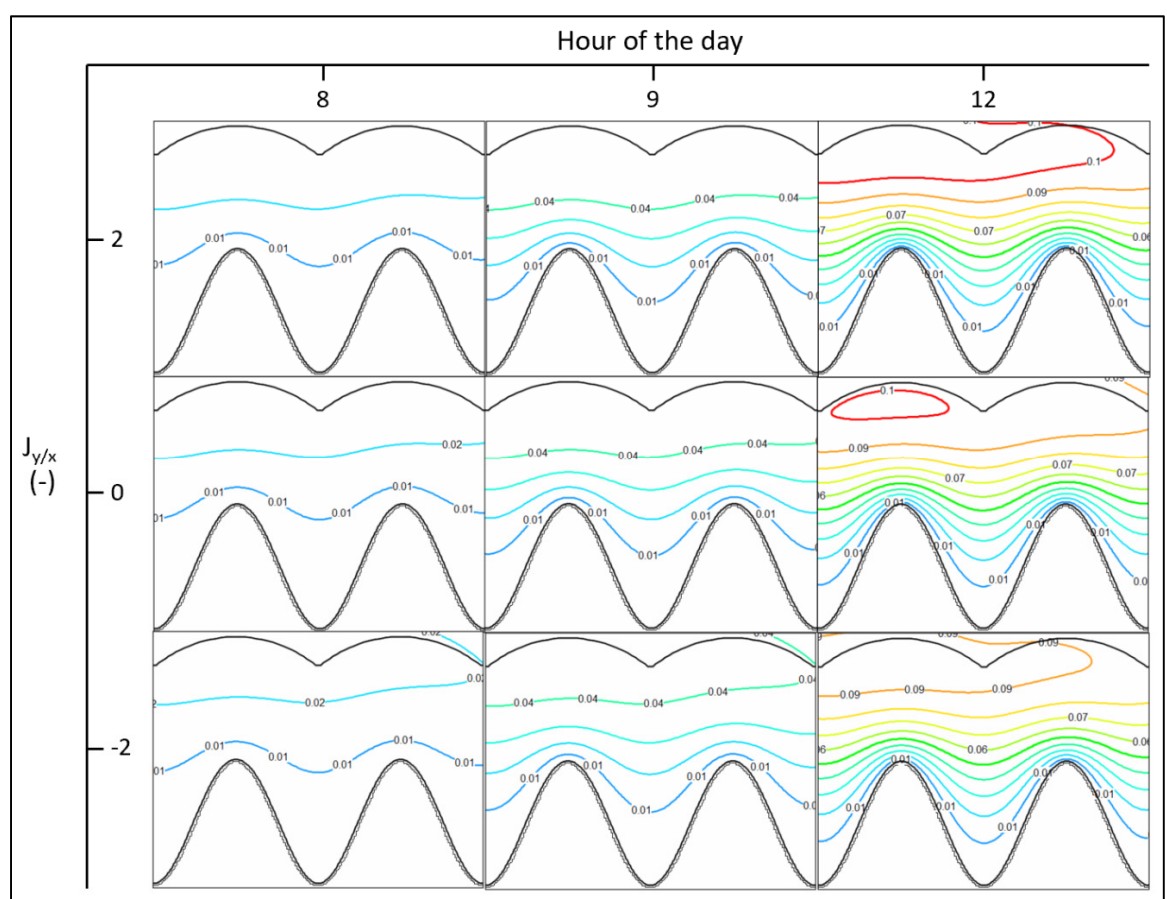

**Figure 9.** Snapshots of drawdown (m) at different h of the day, as a function of regional groundwater flux ($J_{y/x}$; positive for gaining, negative for losing) (rows). All snapshots came from simulations with $ET_{max}$ values of 80 mm/day. Drawdown values are relative to the head values from the first h of the day of each respective simulation.

**Table 3.** Contour line orientations and average head gradients for all regional groundwater flux ($J_{y/x}$) scenarios at ET80. All delta values are 12th h conditions −6th h conditions.

| $J_{y/x}$ | Head Contour Azimuth, ° | | | Head Gradient, % | | |
|---|---|---|---|---|---|---|
| | 6th h | 12th h | Δ | 6th h | 12th h | Δ |
| 2 | 117 | 82 | 35 | 0.115 | 0.179 | 0.064 |
| 1 | 117 | 75 | 42 | 0.114 | 0.213 | 0.099 |
| 0 | 96 | 70 | 26 | 0.125 | 0.253 | 0.128 |
| −1 | 65 | 65 | 0 | 0.185 | 0.307 | 0.122 |
| −2 | 65 | 65 | 0 | 0.256 | 0.376 | 0.120 |

The reoriented head contours formed unique patterns depending on the $J_{y/x}$ scenario. In gaining and neutral scenarios, the contours formed a trough of lower heads parallel to the main axis of the stream; losing scenario contours did not form a trough but compressed towards the stream without a major change in their orientation. When ET = $ET_{max}$, the greatest steepening of head gradients was consistently at the apexes of meander bends where the distance between the stream and maximum drawdown values was smallest. In Figures 9 and 10, only ET80 snapshots are shown, since all active ET scenarios displayed similar patterns. Likewise, J + 1 and J − 1 scenarios were omitted because they showed trends like those of J + 2 and J − 2, respectively.

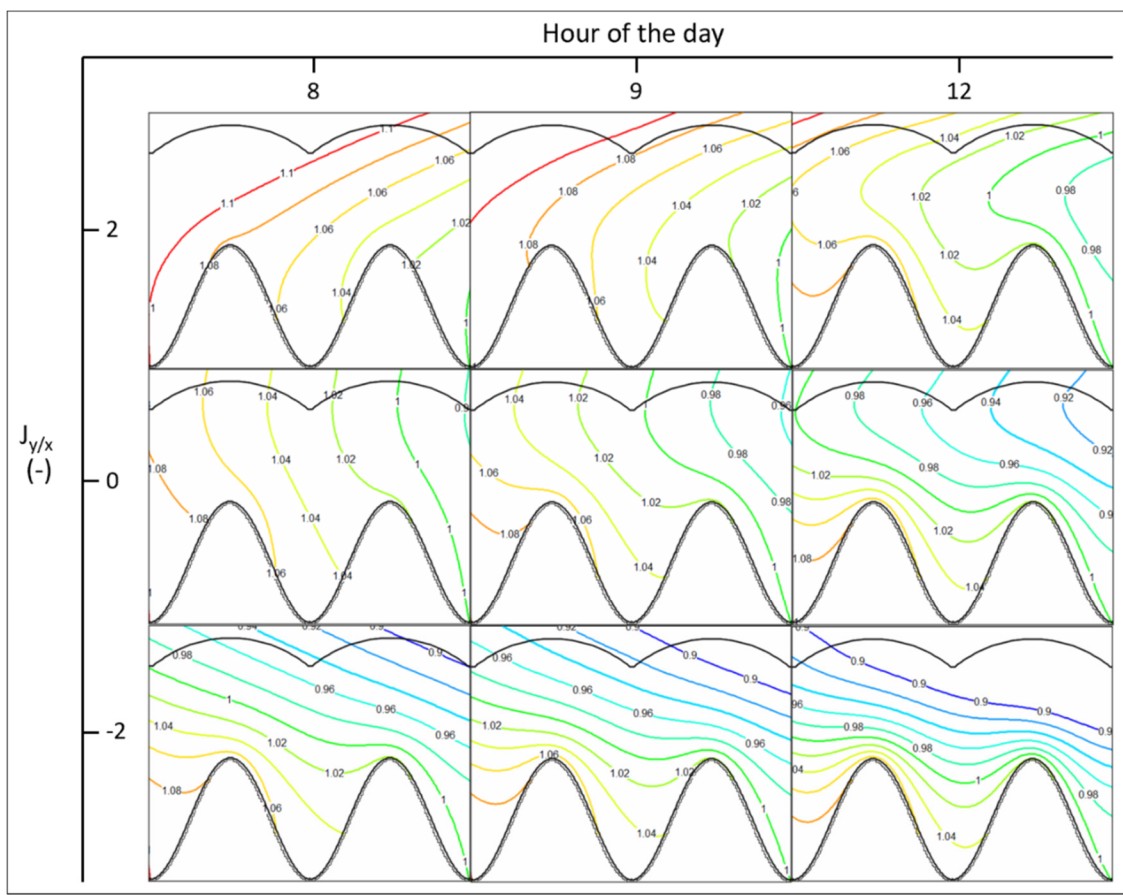

**Figure 10.** Snapshots of head (m) at different h of the day, as a function of regional groundwater flux ($J_{y/x}$; positive for gaining, negative for losing) (rows). All snapshots came from simulations with $ET_{max}$ values of 80 mm/day.

### 3.5. Aquifer Sensitivity

For all model simulations, the aquifer sensitivity parameter Γ was constant at 1.9. This means the duration of effects produced by a daily cycle of ET was almost twice as long as the daily cycle itself. With Γ > 1, perturbations in the water table from ET were not expected to significantly affect the aquifer at sub-daily timescales [28].

## 4. Discussion

### 4.1. Hyporheic Zone Area

The patterns of changes in HZ area sizes and shapes can be explained as the result of active ET altering head distributions and flow paths. In gaining and neutral scenarios, ET temporarily reoriented groundwater flow paths away from the stream and towards the trough of low heads that developed. This means groundwater was flowing away from the stream for a period of h every day, compared to ET0 scenarios where flow paths toward the stream were constant. This change was what gave gaining and neutral HZs larger areas growing primarily in the *y*-direction when ET was active. Losing scenarios experienced much smaller changes in the orientation of head contours, and so their HZs did not undergo much growth or change in shape.

The width of the vegetated belt (30 m) also had a strong influence on HZ geometry because it controlled how head distributions were reoriented. The vegetated belt, represented in this study by the cells with active ET, was wide enough to reach into the alluvial valley well beyond the edge of the stream. This was where troughs of low head values formed in gaining and neutral scenarios, and

therefore where stream water particles were drawn during h of the day with high ET rates. Preliminary tests of the model domain, not detailed in this study, found that a much narrower vegetated belt led to smaller $A_{HZ}$ values relative to no-ET scenarios. The preliminary model had identical values for all the model constants listed in Table 1, but a vegetated belt width of only 2 m. Water table depressions from these ET cells were focused in a narrow band close to the edge of the stream, and stream water particles entering the aquifer tracked more closely to the stream and exited the aquifer more quickly than without ET.

With the addition of ET, HZ areas grew on a scale comparable to the changes produced in other studies by varying basic geomorphic and aquifer parameters and/or adding a dynamic perturbation like ET. In identically-shaped ($S = 1.87$, $\lambda = 40$ m) intrameander zones, Cardenas [12] reduced HZ areas by about 80% when increasing and decreasing $J_{y/x}$ (J0 base case, increased to J + 2 and decreased to J − 2). At higher $ET_{max}$, the neutral and gaining scenarios in this study increased the HZ area close to or surpassing that amount. Gomez-Velez et al. [29] mapped the growth of HZs in response to flood pulses across a range of aquifer sensitivities (Γ values). Comparing those maps to the growth seen in this study, the results of gaining and neutral scenarios were similar to what Gomez-Velez et al. [29] produced in less sensitive ($1 < Γ < 10$) simulations.

With $Γ > 1$ for this study, $A_{HZ}$ was insensitive to hly changes in ET because the hydraulic signal produced by ET took much longer than one h to propagate through the area of interest (i.e., $t_h > 1$ h). Keeping $J_{y/x}$ constant, it is also clear the HZ did respond to changes in $ET_{max}$ between simulations. In the field, $ET_{max}$ is primarily dependent on the intensity of solar radiation, especially in energy-limited environments like riparian zones [27,38]. This is also true for the h-to-h changes in ET withdrawal rates. Daily maximum solar radiation follows long-term cyclic (seasonal) fluctuations [38,39], so it follows that active riparian ET could affect $A_{HZ}$ on primarily seasonal timescales.

## 4.2. Net Groundwater Flux

Like $A_{HZ}$, net flux in the $y$-direction ($Q_y$) and normalized dimensionless flux ($F$) were dependent on the head patterns that developed with active ET. Active ET rearranged head contours so that more water was drawn from the stream into the aquifer, regardless of whether the simulation was originally gaining or losing. All profiles of discrete $Q_y$ values (Figure 4, bottom) captured this: values decreased across the profile, and gaining scenarios showed some cells reversing from positive to negative. When ET = $ET_{max}$, the divide between positive and negative $Q_y$ cells coincided with the troughs of low head values that developed in gaining scenarios.

In gaining scenarios with lower $ET_{max}$, $F$ values dropped below 1 because daily average flux from the aquifer to the stream decreased more than daily average flux from the stream to the aquifer increased (Figure 5). At higher ET rates, both trends continued, but eventually, the decrease in flux to the stream was outpaced by increases in flux to the aquifer, resulting in more total exchange flux than at ET0 ($F > 1$). Looking at gaining scenarios only, the ET rates at which $F$ dropped below 1 correspond to the $ET_{max}$ values for simulations where head contours either did not reorient themselves beyond the $x$-axis or did so only briefly. In other words, even during $ET_{max}$ in the ET20 and ET40 gaining simulations, head contour trendlines never produced an azimuth below 80 degrees (Table 3). Neutral and losing scenario $F$ values only increased with ET because their head gradients did not reorient as strongly or at all in response to ET. These results support the idea that increases in $F$ were due to increases in exchange flux from the stream to the aquifer. The process captured here by which ET increased net flux to the aquifer has been observed in field studies of lateral hyporheic exchange [23,25], suggesting the ET in this model functioned in a way comparable to conceptual models developed from field data.

Increasing ET did draw more water into the ET zone from the alluvial valley, but it was less than the water drawn in from the stream. Comparing ET0 results with ET80 results for all $J_{y/x}$ scenarios, active ET produced an average 0.024% increase in volumetric flux from the alluvial valley, measured at

the A location cell in the $Q_y$ profiles. Using the same scenarios, active ET produced an average 0.17% increase in flux from the stream, measured at the A′ location cell.

The flux metrics developed in previous modeling studies focused on the exchange at the stream–aquifer interface, and therefore can be more easily compared to $F$ than $Q_y$. In general, previous flux metrics were a variation on a Darcy flux term, where the total or average volumetric flux entering or leaving the HZ was divided by the surface area of the stream–aquifer interface. Some of these Darcy fluxes were then made dimensionless by dividing by hydraulic conductivity and two characteristic length terms (see Equation (2)). Each study depicted a slightly different hyporheic exchange flux; but the patterns the studies observed in response to changes in stream morphology, aquifer characteristics, and transient perturbations can still be compared to the effects of ET in this study.

In Cardenas [12], a dimensionless hyporheic flux term was established by dividing the total Darcy flux entering the HZ by hydraulic conductivity. This made the term insensitive to changes in flux leaving the HZ, which this study's $F$ term accounted for. Since the dimensionless values were then normalized to neutral ($J_{y/x} = 0$) scenarios, however, it still describes how a part of the hyporheic exchange flux would respond to changes in ambient groundwater flow. Cardenas [12] showed exponential decreases in their flux term with increases in $|J_{y/x}|$. At $|J_{y/x}| = 2$ and $S = 1.87$, flux was about 70% reduced; in this study, changes in $F$ ranged from -15% to +65%.

Gomez et al. [14] developed two interrelated metrics: a dimensionless Darcy flux normal to the stream, and a dimensionless volumetric exchange flux per unit stream length. Both metrics were used to describe flux leaving the HZ and entering the stream. Since ambient groundwater flow was kept neutral ($J_{y/x} = 0$) in this model, these flux terms captured all exchange at the stream–aquifer interface on the downstream half of the meander bend of interest, and this outgoing flux was symmetrical to the incoming flux on the upstream half of the meander bend. As stream sinuosity (measured here as the ratio of meander amplitude to wavelength) was increased from 1/16 to 6/16, the maximum dimensionless Darcy flux increased over 150% and volumetric exchange flux increased 200% [14]. Neutral scenarios from this study with a similar sinuosity (15/40, the same ratio as 6/16) achieved a maximum increase in $F$ of about 65%. It is likely this study's simulations produced localized changes in exchange flux higher than 65%, since $F$ values were based on flux magnitudes averaged across 24 h and two full stream wavelengths, unlike the metrics used in Gomez et al. [14].

Gomez-Velez et al. [29] established separate metrics for dimensionless fluxes entering the HZ and those leaving the HZ, measured along sections of the stream where the net movement of water was out of and into the stream, respectively. The difference between the fluxes became the net flux leaving the HZ, which was then made dimensionless. In response to a flooding event, dimensionless net flux first decreased and turned negative as stream stage rose and pushed water into the aquifer. Once the stream stage dropped, head gradients reversed and the net flux term increased [29]. These net flux values were not normalized, but all modeled flooding events started at a neutral base case where net flux = 0. Net flux and the total flux used in this study could give very different results for the same scenario, if that scenario had ≈ 0 net flux. The maximum $F$ values in this study were from a scenario comparable to the minimum net flux values produced in Gomez-Velez et al. [29] because they both represented moments where the majority of exchange flux was moving from the stream to the HZ, meaning the net flux /≈ 0. When directly comparing the total dimensionless flux in this study (before being normalized to ET0 scenarios; values not shown) to the minimum dimensionless flux values in Gomez-Velez et al. [29] from simulations with comparable Γ, ET produced responses an order of magnitude lower than those produced by flood events.

## 4.3. Residence Time

The patterns of flux-weighted RTDs can be explained by comparing them to the changes seen in spatial distributions of $RT^*$ (Figure 8). Because these $RT^*$ values were flux-weighted, they reflect changes in a combination of particle travel times and spatially-varying flux at the stream–aquifer interface. Higher $RT^*$ values were a result of longer particle travel times in the area of interest,

higher flux entering the aquifer where and when the particle was released, or a combination of both; the opposite is true for lower $RT^*$ values. The strong early mode observed in gaining scenarios was produced by stream water originating near meander apexes. The shape of these RTDs did not fundamentally change from ET0 to ET80 because active ET did not alter the general shape of $RT^*$ spatial distributions. For neutral scenarios, the early mode in the ET0 RTD came primarily from stream water close to $\sigma^* = 1$. With the addition of active ET, $RT^*$ values for that section of the stream–aquifer interface remained largely unchanged, but the $RT^*$ values further upstream increased and eventually formed a second mode much higher than the first. The average change in $RT^*$ from ET0 to ET80 where $\sigma^* \geq 0.8$ was 19.9; the same average change where $\sigma^* < 0.8$ was 135.9. In losing scenarios, early modes in the RTDs for ET0 came from stream water close to $\sigma^* = 1$ and any later secondary modes present represented water from further upstream. Adding and then intensifying active ET flattened out spatial distributions of $RT^*$, shifting all modes toward the median and reducing standard deviations.

The trends of median $RT^*$ values also correspond to changes in spatial distributions of $RT^*$. Neutral scenarios produced the largest increases in median $RT^*$ because increases in $ET_{max}$ led to larger $RT^*$ for all stream-origin water. For weakly gaining and losing scenarios, increasing $ET_{max}$ caused some sections of $\sigma^*$ to produce lower $RT^*$, but those decreases were outpaced by increased $RT^*$ elsewhere along $\sigma^*$, resulting in small but consistent increases in median $RT^*$. In strongly gaining and losing scenarios, the opposite was almost always true: decreases in $RT^*$ for some stream-origin water outpaced increases in $RT^*$ for other stream-origin water. The sole exception can be seen in J + 2 results when moving from ET60 to ET80. Here, $RT^*$ values from $\sigma^* < 0.3$ had already dropped to ~0 by ET60, but $RT^*$ values from $\sigma^* > 0.3$ still substantially increased.

Previous studies of RTDs in lateral HZs have used mean [29] and mode [13,14] RTs as characteristic timescales for further comparison and analysis. Their choices reflect the model domains that produced particular RTD shapes as well as the subsequent analyses performed. This study used (flux-weighted, dimensionless) median RT values so comparisons to timescales in other studies could be made with the understanding that half of the particles modeled in this study were reaching or exceeding some RT. This is a straightforward way to conceptualize whether simulations were allowing HZ water to reach the timescales necessary for biogeochemical transformation reactions.

Boano et al. [13] produced strongly bimodal probability distributions of flow paths generated in intrameander zones of increasing sinuosity, the lowest sinuosity being higher than the $S = 1.87$ used in this study. The two modes of each distribution reflected different travel times at the neck and apex of meander bends, with the difference between them increasing with sinuosity. These modes were then used as minimum and maximum RTs and compared to timescales for the biogeochemical transformation of organic carbon in order to map the resulting chemical zonations of the HZ. Increasing their meander sinuosity from 2.5 to 5.0, Boano et al. [13] decreased their minimum RT mode by about 200% and increased their maximum RT mode by about 140%. As mentioned in Section 3.3, this study produced maximum increases in median $RT^*$ of about 20% when comparing ET0 scenarios to their respective ET80 scenarios.

Building on this work, Gomez et al. [14] compared their own RTDs to the biogeochemical timescales proposed by Boano et al. [13], but within a model based on the domain used in Cardenas [11] and this study. Gomez et al. [14] chose the first and primary mode of their RTDs as a characteristic timescale. A dimensionless version of the characteristic mode ($\tau^*$) increased as sinuosity increased and as dispersivity decreased. In response to increasing $ET_{max}$, median $RT^*$ increased at a rate comparable to $\tau^*$ responding to sinuosity, but at a much lower rate than $\tau^*$ responding to dispersivity. In Gomez et al. [14], modes other than the characteristic mode also shrank and disappeared from probability distributions with higher sinuosity and lower dispersivity. Increasing $ET_{max}$ was demonstrated to have a similar effect for J − 2 scenarios. Based on characteristic timescales developed in Gomez et al. [14] by applying the definitions of Boano et al. [13] to the data of Zarnetske et al. [14], the simulations in this study produced median $RT^*$ values below the threshold timescales for the transformation of organic carbon ($t_{DOC} = 2.10$ days) and nitrate ($t_{NO3} = 0.92$ days), but above the threshold timescale for oxygen ($t_{O2} = 0.20$

days), discounting median *RT\** values from J − 2 scenarios. *RT\** values are dimensionless, but since they were made dimensionless by dividing RTs of h by a characteristic 24 h, they are equivalent to RTs in units of days.

Gomez-Velez et al. [29] tracked the dimensionless mean RT of all flow paths discharging to the stream from the HZ at a given time during and after flood events. These dimensionless values were normalized to mean RTs from base flow conditions, and these normalized dimensionless values were plotted across time to show relative changes in mean RT as flood pulses interacted with the HZ. For this study, a similar plot of RT changes through time was unnecessary because *RT\** values demonstrated almost no change in response to varying release time for water particles. Even in simulations with comparable Γ values, the RTs from Gomez-Velez et al. [29] were variable in response to flooding whereas RTs from this study did not respond to ET. This difference in response may be because a single daily cycle of ET drew much less water into the aquifer than a flood simulated with similar Γ values. The greater exchange flux from the flood could penetrate farther into the alluvial valley and subsequently take much longer to return to the stream. In some simulations, Gomez-Velez et al. [29] had a part of the HZ detach itself from the intrameander zone and travel down-valley, similar to how particles were pulled into the trough of depression in neutral scenarios of this study. Since this study's particle flow paths did not change if the daily cycle of ET withdrawal rates remained constant, neutral field conditions with a similar aquifer sensitivity could possibly provide a steady supply of stream-origin water to the alluvial valley beyond the intrameander zone. Even outside the geochemical limit of the HZ, this could enhance reaction potential in areas of the aquifer further from the stream by providing aquifer environments reactants unique to the stream.

*4.4. Model Assumptions and Limitations*

The MODFLOW and related models used in this study made assumptions about aspects of the hyporheic system that have been demonstrated to affect hyporheic form and function. The assumptions were made either to keep the study focused on the addition of the dynamic ET sink, so the results could clearly answer the primary research question; or they were made due to the limited time available to run, process, and analyze additional scenarios.

The stream modeled here had an idealized channel morphology, but most of its major characteristics were representative of what can be found in both natural and engineered channels. The stream was assumed to be perfectly sinusoidal, with a constant sinuosity of 1.87 extending indefinitely both upstream and downstream of the domain. Natural channel meanders are often asymmetrical and irregular because of fluctuations between bends in meander wavelength and radius of curvature [45]. Even so, sinusoidal channels can approximate symmetrical and regular meanders produced naturally [45] as well as those engineered to promote meandering and intrameander HZs [46]. Variation in sinuosity has been demonstrated to control hyporheic exchange fluxes [11,14], RTs [13,14], and potential for biogeochemical transformation [13]. Sinuosity was kept constant in order to limit the number of simulations, and a constant value was chosen that fell within the ranges of *S* explored by studies modeling similar domains [11,14,19,29], for ease of comparison. A sinuosity on the higher end of these ranges was chosen to represent a possible situation where a stream is engineered to enhance meandering, and thereby enhance hyporheic exchange flux [47]. Models of slow, meandering, high-order streams from Gomez-Velez and Harvey [48] indicate they produce higher RTs in lateral HZs, and therefore have enhanced potential for biogeochemical processing through lateral exchange compared to higher-relief and lower-order streams.

Taken together, the static aquifer parameters (Table 1) describe a homogeneous aquifer of sand to silty sand. These parameter values were chosen to remain consistent with the ranges of values explored in previous studies [11,13,14,19,29], for ease of comparison. If the model had been designed to represent a gravel aquifer by increasing hydraulic conductivity, specific yield, and porosity, it is possible the time constant Γ would have dropped below 1 and the aquifer would have been more sensitive to hly changes in ET. Studies exploring the effects of median grain size on lateral hyporheic

exchange indicate that a gravel aquifer also would have likely produced higher exchange fluxes at the stream–aquifer interface [12,48].

Some parameters of the model domain were made homogeneous to reduce the number of parameters altered between simulations, but are known to be widely heterogeneous in the field. Assuming the aquifer was completely homogeneous ignored the effects of common paleochannels in intrameander zones, where they have been observed to significantly alter RTDs by providing preferential flow paths for water and solutes [49]. Riverbed and bank sediments in the stream–aquifer interface can hold and transmit water very differently than surrounding aquifer sediment [50]. Since the stream was the only source of water in this model, changing the basic characteristics of this liminal boundary could have strongly influenced the flux of stream water to the aquifer both overall and between hly time steps. Altering the model to represent these elements as heterogeneous would have made results more realistic, but also would have added a complex parameter (degree of heterogeneity) to be accounted for when attempting to interpret the effects of ET on this model.

The boundary conditions of this model were central to its representation of the stream–aquifer–ET system, and some of the assumptions of those boundary conditions limited the realism and variety of situations explored. The model stream, represented with MODFLOW's CHD package, maintained static head despite the adjacent ET sink. This implies the stream was large enough to act as an infinite supply of water to the aquifer. If stream heads had been allowed to fluctuate in response to drawdown in the vegetated zone, the feedback of hydraulic signals could have produced more complex patterns of results like those seen in Gomez-Velez et al. [29]. The static CHD values enforced quasi-steady-state regional groundwater fluxes (RGFs) throughout the rest of the model domain. Fluxes between cells (and by extension, heads) could vary between timesteps, but still generally oscillated around average values, collectively producing a simulation-average water table. These quasi-steady-state groundwater tables helped clarify the hydraulic signals produced by transient ET, but also preclude modeling long-term dynamic equilibrium trends such as regional water table decline and the subsequent response of phreatophyte roots. Long-term decline of average groundwater levels would need to be modeled across a series of simulations where all CHD boundary conditions were altered to enforce new RGF regimes that produced lower water tables.

Many of the assumptions made about the EVT boundary simplified the physiology of the riparian vegetation being modeled. The extinction depth of 2 m may not be appropriate to represent many riparian phreatophyte species, at least in part because of the group's wide variability in extinction depths [42]. Considering that Carroll et al. [42] reported 90% confidence interval values (0.3 to 9 m), no single value would have been fully representative, and 2 m appears to be plausible for riparian phreatophyte species with shorter rooting depths. The model also assumed the phreatophytes were evenly spaced in the vegetated zone and withdrew water at an identical rate set for all EVT cells, before the variation due to head differences. Even spacing is possible when a restored intrameander zone is first being planted, but many field conditions will deviate. Spatial variations in ET could make patterns of drawdown and head less uniform, leading to less predictable flow paths and RTDs.

The EVT package also produced ET withdrawal rates that varied linearly with head. As Shah et al. [41] indicate, an exponential decay relationship more closely matches field observations of ET withdrawal for forested cover. Unlike the EVT package, the Evapotranspiration Segments (ETS) package can have its formula adjusted to account for the exponential and otherwise nonlinear relationships between water table depth and parameters such as $S_y$, the fraction of vadose zone contribution to ET, and rooting density [41]. The EVT package cannot adjust its linear decay of ET and therefore cannot represent these parameters with the same degree of realism. Despite this, the EVT package is adequate for its purpose in this work. The low-order approach of the model included simplifications to the simulation of ET and basic aquifer characteristics, to the point that parameters the ETS package could more accurately represent were already simplified or did not apply. For example, the model aquifer's $S_y$ was constant at all depths; MODFLOW models only saturated flow and therefore could not capture the depth-dependent contribution of the vadose zone to ET; and the root network density of hyporheic

trees was assumed to be completely homogeneous with depth. The EVT package still captured the essential aspect of the relationship between ET and water table depth: a model cell achieved $ET_{max}$ at depth = 0, ET = 0 at the extinction depth, and some fraction of $ET_{max}$ in-between.

As was noted in Section 2.2, the most extreme ET withdrawal rates produced in the ET80 scenarios lie outside what is physically possible in most climate zones [26,27,38,39]. This limits the usefulness of some results in understanding the field conditions of sinuosity-induced hyporheic flow. However, results from ET80 scenarios still contribute to a more thorough exploration of the relationship between the magnitude of ET perturbations and the magnitude of hyporheic zone response. This exploration is central to the goal of this work stated in Section 1.

The temporally symmetrical results for head, drawdown, and flux may be due in part to the assumption of a symmetrical ET curve (Figure 2). This curve was based on the hydrograph used to control the flood stage in Gomez-Velez et al. [29], but it was also a fair approximation of ET estimated in the field [51] and calculated in models with ideal "clear sky" conditions [38]. Despite this, all field estimates of ET reveal day-to-day variation in $ET_{max}$ due to fluctuations in solar radiation [26,38,51], or deviations from the clear sky ideal. Including this variation in the diel ET schedule could be crucial for accurately modeling ET in areas with high cloud cover.

## 5. Conclusions

Hyporheic zones provide unique and critical environments for the mixing of stream water and groundwater. The degree of that mixing depends on static characteristics of the HZ and dynamic changes to sinks and sources of water, such as riparian ET. We explored the effects of near-stream ET in a numerical model of lateral hyporheic exchange. The primary question for this study was: will the introduction of dynamic ET to a numerical hyporheic model diminish HZ area, hyporheic exchange flux, and RTs?

Across the full range of maximum daily ET pumping rates and ambient groundwater orientations explored, ET increased HZ area and hyporheic exchange flux. RTs increased in neutral and weakly gaining and losing conditions but decreased under stronger gaining and losing conditions. Some of these changes were noticeable even at lower ET intensities: HZ areas grew substantially at ET20; areas of the intrameander zone reversed $Q_y$ orientations in gaining scenarios at ET20; and median $RT^*$ decreased by over 20% in strongly gaining scenarios from ET20 to ET40. The changes produced by ET in HZ area and hyporheic exchange flux were comparable to the scale of changes produced in similar modeling studies by altering aquifer characteristics, varying geomorphology, or introducing another time-varying disturbance; but only at higher ET intensities, such as ET60 and ET80. The response of RTs to ET was consistently smaller than the response produced in other studies, in some cases by orders of magnitude. This suggests that at lower ET intensities riparian ET could still be useful in productively altering HZs, but may play a secondary role in HZ modeling when compared to the effects of flood pulses [29,31].

The central mechanism for these responses was the reorientation of local minimum head values away from the stream, resulting in increased flux from the stream to the aquifer. The response of each model simulation to ET depended largely on whether ambient groundwater flow did or did not compliment these locally reoriented head gradients. Gaining scenarios demonstrated the greatest growth in HZ area; losing scenarios showed the greatest increases in *F*; and effects on RT were mixed, but neutral RTDs were altered the most. This study indicates ambient groundwater flow exerts a strong control on the hyporheic response to ET, but whether it enhances or diminishes those responses depends on the desired effect.

The results of this study should be taken with the understanding that some of the underlying assumptions of the model limit its applicability to real stream restoration scenarios. For the restoration of large, low-relief streams specifically, the addition of a wide belt of near-stream phreatophytes can enhance HZ area, RTs, and exchange flux, provided local $J_{y/x}$ stays below 1. Establishing an accurate model of ambient groundwater flow and calculating aquifer sensitivity are crucial to predicting how ET

will affect the HZ in the long-term. For example, this model was effective at preserving the geometry of the HZ because of the low aquifer sensitivity to ET, but results from Gomez-Velez et al. [29] suggest that if sensitivity had been higher ET would have increased fluxes of stream water into the HZ, potentially by an order of magnitude. Some of the positive effects of ET were only achieved at higher ET rates, which may be impossible to achieve in climate zones with low potential ET.

Future models of sinuosity-induced hyporheic exchange should explore the consistency of effects produced by ET across a wider variety of conditions for the aquifer, stream, and ET sink. This work included the Γ sensitivity term from Gomez-Velez et al. [29] to highlight the importance of describing model sensitivity, but it was not the primary focus of the study. A global sensitivity analysis of Γ with respect to the major metrics of this study would simplify the relationships of model parameters while further investigating whether riparian ET as it is currently modeled produces reasonable results. Replacing the static stream with a dynamic, head-dependent flux boundary would allow the model to represent streams small enough to have their stage lowered by nearby ET. Stream stage fluctuations would directly affect fluxes of stream water to the aquifer, which was the primary mechanism for changes to the HZ in this study. Introducing spatial heterogeneity into aquifer characteristics and day-to-day fluctuations in the ET schedule would likely provide more realistic results that can inform future efforts to restore the ecosystem services of lateral hyporheic zones.

**Supplementary Materials:** The following are available online at http://www.mdpi.com/2073-4441/12/2/424/s1, a separate Supporting Information document ("SupportingInformation.docx") provides detailed descriptions of some processes undertaken in the Methods section, to make it easier to recreate the model and the results produced. Copies of the ModelMuse project file, model inputs and outputs, and novel pre- and post-processing code used in this research have been organized for public release. They will be made available by being uploaded as supplementary information to the publication.

**Author Contributions:** Conceptualization, all authors; methodology, all authors; software, J.K. and T.A.E.; validation, J.K. and T.A.E.; formal analysis, J.K. and T.A.E.; investigation, J.K. and T.A.E.; resources, all authors; data curation, J.K. and T.A.E.; writing—original draft preparation, all authors; writing—review and editing, all authors; visualization, all authors; supervision, T.A.E., J.G.-V. and L.K.L.; project administration, J.K. and T.A.E. All authors have read and agreed to the published version of the manuscript.

**Funding:** This research received no external funding. J.G.-V. is funded by the U.S. National Science Foundation (award EAR 1830172) and U.S. Department of Energy (DOE), Office of Biological and Environmental Research (BER), as part of BER's Subsurface Biogeochemistry Research Program (SBR). This contribution originates from the SBR Scientific Focus Area (SFA) at the Pacific Northwest National Laboratory (PNNL).

**Acknowledgments:** Special thanks to Richard B. Winston, Ph.D., for his technical support and guidance in using MODFLOW and ModelMuse via written correspondence.

**Conflicts of Interest:** The authors declare no conflicts of interest.

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
