# Peer review of "Dynamic Evapotranspiration Alters Hyporheic Flow and Residence Times in the Intrameander Zone"

_water, doi:10.3390/w12020424_

Round 1

Reviewer 1 Report

Thank you for inviting me to review this manuscript. This is an interesting manuscript that attempts to understand the water exchange among stream-aquifer-ET systems. The paper was written very well, and the question was posed clearly. However, I have three major comments.

First, authors set five different scenarios for ET withdrawal rates, from 0 to 80 mm/day (Lines 207-210). I questioned the ET of 80 mm/day. According to studies in hyper-arid of China, where widely distributes the woody phreatophytes (e.g., Wang, P. et al., 2014. Journal of Hydrology, 519, Part B(0): 2289-2300; Yuan et al, 2014. Agricultural and Forest Meteorology, 194(0): 144-154.), the annual potential evaporation is only approximately 1500 mm. 80 mm/day is too large.

Second, linear relationship between depth and ET0 as described by ETV package is not the best choice. I suggest to use Evapotranspiration Segments (ETS) package instead of ETV. The extinct depth was set in the manuscript is not convinced. I strongly recommend the following paper to authors.

Shah, N., et al., 2007. Extinction depth and evapotranspiration from ground water under selected land covers. Ground Water, 45(3): 329-338.

Third, in lines 680-693, authors discussed the representation of plant roots for woody phreatophytes. The question is how the roots dynamically interact with groundwater table changes? I also recommend two following papers.

Fan, Y., et al., 2017. Hydrologic regulation of plant rooting depth. Proceedings of the National Academy of Sciences, 114(40): 10572-10577. DOI:10.1073/pnas.1712381114

Wang, P. et al., 2018. Implementing Dynamic Root Optimization in Noah‐MP for Simulating Phreatophytic Root Water Uptake. Water Resources Research, 54(3): 1560-1575. DOI:doi:10.1002/2017WR021061

Reviewer 2 Report

Dear authors,

I read with interest your paper entitled “Dynamic Evapotranspiration Alters Hyporheic Flow and Residence Times in the Intrameander Zone”. The paper deals with a synthetic study investigating the role of evapotransporation on the hydrogeological behavior of an aquifer in equilibrium with a meandering river. The paper first introduces the objective of the study and review relevant literature on the topic. It then clearly describes the parameters that will be used to assess the effect of evapotranspiration rate and groundwater gradient on the hyporheic flow. The significant and representative results are selected, thoroughly analyzed and discussed.

Overall, I find the paper very interesting, well written and scientifically sound and recommend its publication. I only have very minor remarks/suggestions to broaden the discussion and open perspectives for future work:

There is only one subsection 1.1 in the introduction. I would therefore upgrade this part to a full section “2. Overview of previous modelling effort” Some clarification about the transport model could be useful. It is currently spread at different locations (L174-176, L227-233, 233-243). If I understand correctly, you run a steady-state MODFLOW simulation with no evaporation, then a MT3DMS simulation with a fixed concentration of 1 g/m3 in the river still without EVT, at the end of which you run the transient simulation with EVT and then track the particles with MODPATH. It is not clear what concentration you impose during the transient part of the transport (No additional input. Can you clarify (maybe with a table)? I would also indicate on Figure 1 the segment along which the tracer is released. Figure 3. In the caption, it is indicated “The colors on each map correspond to concentrations of stream water that have entered the aquifer; the range of concentrations are different for each simulation and so the colors cannot be compared between maps. The colors follow a rainbow gradient where blue indicates lower concentrations and red indicates higher concentrations.” However, following the definition at line 237, the range of concentration values in the hyporheic zone should always be between 0.5 and 1 g/m³, so that a common colorscale could be used in Figure 3, right ? I don’t really understand the interest to normalize those values as indicated L242-243. The normalization is mostly interesting to calculate the increase/decrease in size, is it the only objective? L473 refers to a preliminary study mentioned in the Introduction section. Please, repeat the reference as I cannot find which preliminary study it refers to. L605-619 and L740-749. Given the high number of parameters potentially influencing the results and interacting together, it would be interesting to perform in the future a global sensitivity to identify the most important parameters (including boundary conditions) and the interaction between parameters (e.g. gradient and hydraulic conductivity will interact for the calculation of the flux). Global sensitivity is computationally expensive, but recent development using distance-based approaches (Park et al., 2016) allow to estimate global sensitivity with a limited number of simulations (typically a few hundreds). See for example recent applications in hydrogeology by (Hermans et al., 2019 and Hoffman et al., 2019). This seems feasible in this context since you already use well-defined model-output (HZ area, fluxes, etc.). L669-672. Not only the heterogeneity of the aquifer, but also the heterogeneity of the riverbed and riverbanks will influence the exchange fluxes between the river and the aquifer. This could be worth mentioning (e.g., Brunner et al., 2017, Ghysels et al., 2019).

References

Brunner P, Therrien R, Renard P, Simmons CT, Franssen HJH. 2017. Advances in understanding river–groundwater interactions. Review of Geophysics, 55(3):818–854.

Ghysels, G., Mutua, S., Baya Veliz, G., Huysmans, M. 2019. A modified approach for modelling river–aquifer interaction of gaining rivers in MODFLO, including riverbed heterogeneity and river bank seepage. Hydrogeology Journal, 27:1851–1863

Hermans, T., Lesparre, N., De Schepper, G., Robert, T., 2019. Bayesian evidential learning: a field validation using push-pull tests. Hydrogeol J 27, 1661–1672. https://doi.org/10.1007/s10040-019-01962-9

Hoffmann, R., Dassargues, A., Goderniaux, P., Hermans, T., 2019. Heterogeneity and prior uncertainty investigation using a joint heat and solute tracer experiment in alluvial sediments. Frontiers in Earth Sciences - Hydrosphere, Parameter Estimation and Uncertainty Quantification in Water Resources Modeling 7, 108.

Park, J., Yang, G., Satija, A., Scheidt, C., Caers, J., 2016. DGSA: A Matlab toolbox for distance-based generalized sensitivity analysis of geoscientific computer experiments. Computers & Geosciences 97, 15–29. https://doi.org/10.1016/j.cageo.2016.08.021

Round 2

Reviewer 1 Report

I appreciate the authors for their detailed response to my comments. The authors stated that "The authors also believe the model’s current extinction depth (2 m, equal to rooting depth) is an adequate representation and can remain unchanged. " I disagree with this statement. At least, for the riparian areas with phreatopytes in the northwest of China, it is not true. Besides, I am also not satisfied with the response to 3rd comment. Globally, the groundwater level is declining in the arid riparian regions. How does the plant transpiration change with dynamic plant roots under the water table declines, this needs to be discussed. All simulations in the study were done, in my option, for quasi-steady assumption. These two suggestions is only for authors' consideration. I thank for their response again, and I will not take reviews further.
